# Hardness of Online Sleeping Combinatorial Optimization Problems

**Satyen Kale**[*][†]
Yahoo Research
satyen@satyenkale.com

**Chansoo Lee**[†]
Univ. of Michigan, Ann Arbor
chansool@umich.edu

**Dávid Pál**
Yahoo Research
dpal@yahoo-inc.com

## Abstract

We show that several online combinatorial optimization problems that admit efficient no-regret algorithms become computationally hard in the sleeping setting where a subset of actions becomes unavailable in each round. Specifically, we show that the sleeping versions of these problems are at least as hard as PAC learning DNF expressions, a long standing open problem. We show hardness for the sleeping versions of ONLINE SHORTEST PATHS, ONLINE MINIMUM SPANNING TREE, ONLINE $k$-SUBSETS, ONLINE $k$-TRUNCATED PERMUTATIONS, ONLINE MINIMUM CUT, and ONLINE BIPARTITE MATCHING. The hardness result for the sleeping version of the Online Shortest Paths problem resolves an open problem presented at COLT 2015 [Koolen et al., 2015].

## 1  Introduction

Online learning is a sequential decision-making problem where learner repeatedly chooses an action in response to adversarially chosen losses for the available actions. The goal of the learner is to minimize the *regret*, defined as the difference between the total loss of the algorithm and the loss of the best fixed action in hindsight. In online combinatorial optimization, the actions are subsets of a ground set of *elements* (also called *components*) with some combinatorial structure. The loss of an action is the sum of the losses of its elements. A particular well-studied instance is the ONLINE SHORTEST PATH problem [Takimoto and Warmuth, 2003] on a graph, in which the actions are the paths between two fixed vertices and the elements are the edges.

We study a *sleeping* variant of online combinatorial optimization where the adversary not only chooses losses but *availability* of the elements every round. The unavailable elements are called *sleeping* or *sabotaged*. In ONLINE SABOTAGED SHORTEST PATH problem, for example, the adversary specifies unavailable edges every round, and consequently the learner cannot choose any path using those edges. A straightforward application of the sleeping experts algorithm proposed by Freund et al. [1997] gives a no-regret learner, but it takes exponential time (in the input graph size) every round. The design of a computationally efficient no-regret algorithm for ONLINE SABOTAGED SHORTEST PATH problem was presented as an open problem at COLT 2015 by Koolen et al. [2015].

In this paper, we resolve this open problem and prove that ONLINE SABOTAGED SHORTEST PATH problem is computationally hard. Specifically, we show that a polynomial-time low-regret algorithm for this problem implies a polynomial-time algorithm for PAC learning DNF expressions, which is a long-standing open problem. The best known algorithm for PAC learning DNF expressions on $n$ variables has time complexity $2^{\widetilde{O}(n^{1/3})}$ [Klivans and Servedio, 2001].

---

[*]Current affiliation: Google Research.

[†]This work was done while the authors were at Yahoo Research.

Our reduction framework (Section 4) in fact shows a general result that any online sleeping combinatorial optimization problem with two simple structural properties is as hard as PAC learning DNF expressions. Leveraging this result, we obtain hardness results for the sleeping variant of well-studied online combinatorial optimization problems for which a polynomial-time no-regret algorithm exists: ONLINE MINIMUM SPANNING TREE, ONLINE $k$-SUBSETS, ONLINE $k$-TRUNCATED PERMUTATIONS, ONLINE MINIMUM CUT, and ONLINE BIPARTITE MATCHING (Section 5).

Our hardness result applies to the worst-case adversary as well as a *stochastic* adversary, who draws an i.i.d. sample every round from a fixed (but unknown to the learner) joint distribution over availabilities and losses. This implies that no-regret algorithms would require even stronger restrictions on the adversary.

## 1.1 Related Work

**Online Combinatorial Optimization.** The standard problem of online linear optimization with $d$ actions (Experts setting) admits algorithms with $O(d)$ running time per round and $O(\sqrt{T \log d})$ regret after $T$ rounds [Littlestone and Warmuth, 1994, Freund and Schapire, 1997], which is minimax optimal [Cesa-Bianchi and Lugosi, 2006, Chapter 2]. A naive application of such algorithms to online combinatorial optimization problem (precise definitions to be given momentarily) over a ground set of $d$ elements will result in $\exp(O(d))$ running time per round and $O(\sqrt{T}d)$ regret.

Despite this, many online combinatorial optimization problems, such as the ones considered in this paper, admit algorithms with[3] poly$(d)$ running time per round and $O(\text{poly}(d)\sqrt{T})$ regret [Takimoto and Warmuth, 2003, Kalai and Vempala, 2005, Koolen et al., 2010, Audibert et al., 2013]. In fact, Kalai and Vempala [2005] shows that the existence of a polynomial-time algorithm for an offline combinatorial problem implies the existence of an algorithm for the corresponding online optimization problem with the same per-round running time and $O(\text{poly}(d)\sqrt{T})$ regret.

**Online Sleeping Optimization.** In studying online sleeping optimization, three different notions of regret have been used: (a) policy regret, (b) ranking regret, and (c) per-action regret, in decreasing order of computational hardness to achieve no-regret. *Policy regret* is the total difference between the loss of the algorithm and the loss of the best policy, which maps a set of available actions and the observed loss sequence to an available action [Neu and Valko, 2014]. *Ranking regret* is the total difference between the loss of the algorithm and the loss of the best ranking of actions, which corresponds to a policy that chooses in each round the highest-ranked available action [Kleinberg et al., 2010, Kanade and Steinke, 2014, Kanade et al., 2009]. *Per-action regret* is the difference between the loss of the algorithm and the loss of an action, summed over only the rounds in which the action is available [Freund et al., 1997, Koolen et al., 2015]. Note that policy regret upper bounds ranking regret, and while ranking regret and per-action regret are generally incomparable, per-action regret is usually the smallest of the three notions.

The sleeping Experts (also known as Specialists) setting has been extensively studied in the literature [Freund et al., 1997, Kanade and Steinke, 2014]. In this paper we focus on the more general online sleeping combinatorial optimization problem, and in particular, the per-action notion of regret.

A summary of known results for online sleeping optimization problems is given in Figure 1. Note in particular that an efficient algorithm was known for minimizing per-action regret in the sleeping Experts problem [Freund et al., 1997]. We show in this paper that a similar efficient algorithm for minimizing per-action regret in online sleeping combinatorial optimization problems cannot exist, unless there is an efficient algorithm for learning DNFs. Our reduction technique is closely related to that of Kanade and Steinke [2014], who reduced agnostic learning of disjunctions to *ranking regret* minimization in the sleeping Experts setting.

## 2 Preliminaries

An instance of online combinatorial optimization is defined by a *ground set* $U$ of $d$ elements, and a *decision set* $\mathcal{D}$ of actions, each of which is a subset of $U$. In each round $t$, the online learner is required to choose an action $V_t \in \mathcal{D}$, while simultaneously an adversary chooses a loss function

| Regret notion | Bound | Sleeping Experts | Sleeping Combinatorial Opt. |
|---|---|---|---|
| Policy | Upper | $O(\sqrt{T \log d})$, under ILA [Kanade et al., 2009] | $O(\text{poly}(d)\sqrt{T})$, under ILA [Neu and Valko, 2014, Abbasi-Yadkori et al., 2013] |
| | Lower | | $\Omega(\text{poly}(d)T^{1-\delta})$, under SLA [Abbasi-Yadkori et al., 2013] |
| Ranking | Lower | $\Omega(\text{poly}(d)T^{1-\delta})$, under SLA [Kanade and Steinke, 2014] | $\Omega(\exp(\Omega(d))\sqrt{T})$, under SLA [Easy construction, omitted] |
| Per-action | Upper | $O(\sqrt{T \log d})$, adversarial setting [Freund et al., 1997] | |
| | Lower | | $\Omega(\text{poly}(d)T^{1-\delta})$, under SLA [This paper] |

Figure 1: Summary of known results. *Stochastic Losses and Availabilities* (SLA) assumption is where adversary chooses a joint distribution over loss and availability before the first round, and takes an i.i.d. sample every round. *Independent Losses and Availabilities* (ILA) assumption is where adversary chooses losses and availabilities independently of each other (one of the two may be adversarially chosen; the other one is then chosen i.i.d in each round). Policy regret upper bounds ranking regret which in turn upper bounds per-action regret for the problems of interest; hence some bounds shown in some cells of the table carry over to other cells by implication and are not shown for clarity. The lower bound on ranking regret in online sleeping combinatorial optimization is unconditional and holds for any algorithm, efficient or not. All other lower bounds are *computational*, i.e. for polynomial time algorithms, assuming intractability of certain well-studied learning problems, such as learning DNFs or learning noisy parities.

$\ell_t : U \to [-1, 1]$. The loss of any $V \in \mathcal{D}$ is given by (with some abuse of notation)

$$\ell_t(V) := \sum_{e \in V} \ell_t(e).$$

The learner suffers loss $\ell_t(V_t)$ and obtains $\ell_t$ as feedback. The regret of the learner with respect to an action $V \in \mathcal{D}$ is defined to be

$$\text{Regret}_T(V) := \sum_{t=1}^{T} \ell_t(V_t) - \ell_t(V).$$

We say that an online optimization algorithm has a regret bound of $f(d, T)$ if $\text{Regret}_T(V) \leq f(d, T)$ for all $V \in \mathcal{D}$. We say that the algorithm has *no regret* if $f(d, T) = \text{poly}(d)T^{1-\delta}$ for some $\delta \in (0, 1)$, and it is *computationally efficient* if it has a per-round running time of order $\text{poly}(d, T)$.

We now define an instance of the online sleeping combinatorial optimization. In this setting, at the start of each round $t$, the adversary selects a set of *sleeping elements* $S_t \subseteq U$ and reveals it to the learner. Define $\mathcal{A}_t = \{V \in \mathcal{D} \mid V \cap S_t = \emptyset\}$, the set of *awake actions* at round $t$; the remaining actions in $\mathcal{D}$, called *sleeping actions*, are unavailable to the learner for that round. If $\mathcal{A}_t$ is empty, i.e., there are no awake actions, then the learner is not required to do anything for that round and the round is discarded from computation of the regret.

For the rest of the paper, unless noted otherwise, we use *per-action regret* as our performance measure. Per-action regret with respect to $V \in \mathcal{D}$ is defined as:

$$\text{Regret}_T(V) := \sum_{t: V \in \mathcal{A}_t} \ell_t(V_t) - \ell_t(V). \tag{1}$$

In other words, our notion of regret considers only the rounds in which $V$ is awake.

For clarity, we define an online combinatorial optimization *problem* as a family of *instances* of online combinatorial optimization (and correspondingly for online sleeping combinatorial optimization). For example, ONLINE SHORTEST PATH *problem* is the family of all *instances* of all graphs with designated source and sink vertices, where the decision set $\mathcal{D}$ is a set of paths from the source to sink, and the elements are edges of the graph.

Our main result is that many natural online sleeping combinatorial optimization problems are unlikely to admit a computationally efficient no-regret algorithm, although their non-sleeping versions (i.e., $\mathcal{A}_t = \mathcal{D}$ for all $t$) do. More precisely, we show that these online sleeping combinatorial optimization problems are at least as hard as PAC learning DNF expressions, a long-standing open problem.

## 3 Online Agnostic Learning of Disjunctions

Instead of directly reducing PAC learning DNF expressions to no-regret learning for online sleeping combinatorial optimization problems, we use an intermediate problem, online agnostic learning of disjunctions. By a standard online-to-batch conversion argument [Kanade and Steinke, 2014], online agnostic learning of disjunctions is at least as hard as agnostic improper PAC-learning of disjunctions [Kearns et al., 1994], which in turn is at least as hard as PAC-learning of DNF expressions [Kalai et al., 2012]. The online-to-batch conversion argument allows us to assume the stochastic adversary (i.i.d. input sequence) for online agnostic learning of disjunctions, which in turn implies that our reduction applies to online sleeping combinatorial optimization with a stochastic adversary.

Online agnostic learning of disjunctions is a repeated game between the adversary and a learning algorithm. Let $n$ denote the number of variables in the disjunction. In each round $t$, the adversary chooses a vector $\mathbf{x}_t \in \{0,1\}^n$, the algorithm predicts a label $\widehat{y}_t \in \{0,1\}$ and then the adversary reveals the correct label $y_t \in \{0,1\}$. If $\widehat{y}_t \neq y_t$, we say that algorithm makes an error.

For any predictor $\phi : \{0,1\}^n \to \{0,1\}$, we define the *regret* with respect to $\phi$ after $T$ rounds as

$$\text{Regret}_T(\phi) = \sum_{t=1}^{T} \mathbf{1}[\widehat{y}_t \neq y_t] - \mathbf{1}[\phi(\mathbf{x}_t) \neq y_t].$$

Our goal is to design an algorithm that is competitive with any disjunction, i.e. for any disjunction $\phi$ over $n$ variables, the regret is bounded by $\text{poly}(n) \cdot T^{1-\delta}$ for some $\delta \in (0,1)$. Recall that a disjunction over $n$ variables is a boolean function $\phi : \{0,1\}^n \to \{0,1\}$ that on an input $\mathbf{x} = (x(1), x(2), \ldots, x(n))$ outputs

$$\phi(\mathbf{x}) = \left( \bigvee_{i \in P} x(i) \right) \vee \left( \bigvee_{i \in N} \overline{x(i)} \right)$$

where $P$ and $N$ are disjoint subsets of $\{1, 2, \ldots, n\}$. We allow either $P$ or $N$ to be empty, and the empty disjunction is interpreted as the constant 0 function. For any index $i \in \{1, 2, \ldots, n\}$, we call it a *relevant index* for $\phi$ if $i \in P \cup N$ and *irrelevant index* for $\phi$ otherwise. For any relevant index $i$, we call it *positive* if $i \in P$ and *negative* if $i \in N$.

## 4 General Hardness Result

In this section, we identify two combinatorial properties of online sleeping combinatorial optimization problems that are computationally hard.

**Definition 1.** *Let $n$ be a positive integer. Consider an instance of online sleeping combinatorial optimization where the ground set $U$ has $d$ elements with $3n + 2 \leq d \leq \text{poly}(n)$. This instance is called a **hard instance with parameter** $n$, if there exists a subset $U_s \subseteq U$ of size $3n + 2$ and a bijection between $U_s$ and the set (i.e., labeling of elements in $U_s$ by the set)*

$$\bigcup_{i=1}^{n} \{(i,0), (i,1), (i,\star)\} \cup \{0,1\},$$

*such that the decision set $\mathcal{D}$ satisfies the following properties:*

1. *(**Heaviness**) Any action $V \in \mathcal{D}$ has at least $n + 1$ elements in $U_s$.*

2. *(**Richness**) For all $(s_1, \ldots, s_{n+1}) \in \{0, 1, \star\}^n \times \{0, 1\}$, the action $\{(1, s_1), (2, s_2), \ldots, (n, s_n), s_{n+1}\} \in U_s$ is in $\mathcal{D}$.*

We now show how to use the above definition of hard *instances* to prove the hardness of an online sleeping combinatorial optimization (OSCO) *problem* by reducing from the online agnostic learning of disjunction (OALD) problem. At a high level, the reduction works as follows. Given an instance of the OALD problem, we construct a specific instance of the the OSCO and a sequence of losses and availabilities based on the input to the OALD problem. This reduction has the property that for any disjunction, there is a special set of actions of size $n + 1$ such that (a) exactly one action is available in any round and (b) the loss of this action exactly equals the loss of the disjunction on the current input example. Furthermore, the action chosen by the OSCO can be converted into a prediction in the OALD problem with only lesser or equal loss. These two facts imply that the regret of the OALD algorithm is at most $n + 1$ times the per-action regret of the OSCO algorithm.

**Algorithm 1** ALGORITHM ALG$_{\text{DISJ}}$ FOR LEARNING DISJUNCTIONS

---

**Require:** An algorithm Alg$_{\text{osco}}$ for the online sleeping combinatorial optimization problem, and the input size $n$ for the disjunction learning problem.
1: Construct a hard instance $(U, \mathcal{D})$ with parameter $n$ of the online sleeping combinatorial optimization problem, and run Alg$_{\text{osco}}$ on it.
2: **for** $t = 1, 2, \ldots, T$ **do**
3:   Receive $\mathbf{x}_t \in \{0, 1\}^n$.
4:   Set the set of sleeping elements for Alg$_{\text{osco}}$ to be $S_t = \{(i, 1 - x_t(i)) \mid i = 1, 2, \ldots, n\}$.
5:   Obtain an action $V_t \in \mathcal{D}$ by running Alg$_{\text{osco}}$ such that $V_t \cap S_t = \emptyset$.
6:   Set $\widehat{y}_t = \mathbf{1}[0 \notin V_t]$.
7:   Predict $\widehat{y}_t$, and receive true label $y_t$.
8:   In algorithm Alg$_{\text{osco}}$, set the loss of the awake elements $e \in U \setminus S_t$ as follows:

$$\ell_t(e) = \begin{cases} \frac{1 - y_t}{n + 1} & \text{if } e \neq 0 \\ y_t - \frac{n(1 - y_t)}{n + 1} & \text{if } e = 0. \end{cases}$$

9: **end for**

---

**Theorem 1.** *Consider an online sleeping combinatorial optimization problem such that for any positive integer $n$, there is a hard instance with parameter $n$ of the problem. Suppose there is an algorithm Alg$_{\text{osco}}$ that for any instance of the problem with ground set $U$ of size $d$, runs in time $\text{poly}(T, d)$ and has regret bounded by $\text{poly}(d) \cdot T^{1-\delta}$ for some $\delta \in (0, 1)$. Then, there exists an algorithm Alg$_{\text{disj}}$ for online agnostic learning of disjunctions over $n$ variables with running time $\text{poly}(T, n)$ and regret $\text{poly}(n) \cdot T^{1-\delta}$.*

*Proof.* Alg$_{\text{disj}}$ is given in Algorithm 1. First, we note that in each round $t$, we have

$$\ell_t(V_t) \geq \mathbf{1}[y_t \neq \widehat{y}_t]. \tag{2}$$

We prove this separately for two different cases; in both cases, the inequality follows from the heaviness property, i.e., the fact that $|V_t| \geq n + 1$.

1. If $0 \notin V_t$, then the prediction of Alg$_{\text{disj}}$ is $\widehat{y}_t = 1$, and thus

$$\ell_t(V_t) = |V_t| \cdot \frac{1 - y_t}{n + 1} \geq 1 - y_t = \mathbf{1}[y_t \neq \widehat{y}_t].$$

2. If $0 \in V_t$, then the prediction of Alg$_{\text{disj}}$ is $\widehat{y}_t = 0$, and thus

$$\ell_t(V_t) = (|V_t| - 1) \cdot \frac{1 - y_t}{n + 1} + \left( y_t - \frac{n(1 - y_t)}{n + 1} \right) \geq y_t = \mathbf{1}[y_t \neq \widehat{y}_t].$$

Note that if $V_t$ satisfies the equality $|V_t| = n + 1$, then we have an equality $\ell_t(V_t) = \mathbf{1}[y_t \neq \widehat{y}_t]$; this property will be useful later.

Next, let $\phi$ be an arbitrary disjunction, and let $i_1 < i_2 < \cdots < i_m$ be its relevant indices sorted in increasing order. Define $f_\phi : \{1, 2, \ldots, m\} \to \{0, 1\}$ as $f_\phi(j) := \mathbf{1}[i_j$ is a positive index for $\phi]$, and define the set of elements $W_\phi := \{(i, \star) \mid i$ is an irrelevant index for $\phi\}$. Finally, let $\mathcal{D}_\phi = \{V_\phi^1, V_\phi^2, \ldots, V_\phi^{m+1}\}$ be the set of $m + 1$ actions where for $j = 1, 2, \ldots, m$, we define

$$V_\phi^j := \{(i_\ell, 1 - f_\phi(\ell)) \mid 1 \leq \ell < j\} \cup \{(i_j, f_\phi(j))\} \cup \{(i_\ell, \star) \mid j < \ell \leq m\} \cup W_\phi \cup \{1\},$$

and

$$V_\phi^{m+1} := \{(i_\ell, 1 - f_\phi(\ell)) \mid 1 \leq \ell \leq m\} \cup W_\phi \cup \{0\}.$$

The actions in $\mathcal{D}_\phi$ are indeed in the decision set $\mathcal{D}$ due to the richness property.

We claim that $\mathcal{D}_\phi$ contains exactly one awake action in every round and the awake action contains the element 1 if and only if $\phi(\mathbf{x}_t) = 1$. First, we prove uniqueness: if $V_\phi^j$ and $V_\phi^k$ (where $j < k$) are both awake in the same round, then $(i_j, f_\phi(j)) \in V_\phi^j$ and $(i_j, 1 - f_\phi(j)) \in V_\phi^k$ are both awake elements, contradicting our choice of $S_t$. To prove the rest of the claim, we consider two cases:

1. If $\phi(\mathbf{x}_t) = 1$, then there is at least one $j \in \{1, 2, \dots, m\}$ such that $x_t(i_j) = f_\phi(j)$. Let $j'$ be the smallest such $j$. Then, by construction, the set $V_\phi^{j'}$ is awake at time $t$, and $1 \in V_\phi^{j'}$, as required.

2. If $\phi(\mathbf{x}_t) = 0$, then for all $j \in \{1, 2, \dots, m\}$ we must have $x_t(i_j) = 1 - f_\phi(j)$. Then, by construction, the set $V_\phi^{m+1}$ is awake at time $t$, and $0 \in V_\phi^{m+1}$, as required.

Since every action in $\mathcal{D}_\phi$ has exactly $n + 1$ elements, and if $V$ is awake action in $\mathcal{D}_\phi$ at time $t$, we just showed that $1 \in V$ if and only if $\phi(\mathbf{x}_t) = 1$, exactly the same argument as in the beginning of this proof implies that

$$\ell_t(V) = \mathbf{1}[y_t \neq \phi(\mathbf{x}_t)]. \tag{3}$$

Furthermore, since exactly one action in $\mathcal{D}_\phi$ is awake every round, we have

$$\sum_{t=1}^{T} \mathbf{1}[y_t \neq \phi(\mathbf{x}_t)] = \sum_{V \in \mathcal{D}_\phi} \sum_{t:\, V \in \mathcal{A}_t} \ell_t(V). \tag{4}$$

Finally, we can bound the regret of algorithm $\mathsf{Alg}_{\mathsf{disj}}$ (denoted $\mathrm{Regret}_T^{\mathsf{disj}}$) in terms of the regret of algorithm $\mathsf{Alg}_{\mathsf{osco}}$ (denoted $\mathrm{Regret}_T^{\mathsf{osco}}$) as follows:

$$\mathrm{Regret}_T^{\mathsf{disj}}(\phi) = \sum_{t=1}^{T} \mathbf{1}[\widehat{y}_t \neq y_t] - \mathbf{1}[\phi(\mathbf{x}_t) \neq y_t] \leq \sum_{V \in \mathcal{D}_\phi} \sum_{t:\, V \in \mathcal{A}_t} \ell_t(V_t) - \ell_t(V)$$

$$= \sum_{V \in \mathcal{D}_\phi} \mathrm{Regret}_T^{\mathsf{osco}}(V) \leq |\mathcal{D}_\phi| \cdot \mathrm{poly}(d) \cdot T^{1-\delta} = \mathrm{poly}(n) \cdot T^{1-\delta},$$

The first inequality follows by (2) and (4), and the last equation since $|\mathcal{D}_\phi| \leq n + 1$ and $d \leq \mathrm{poly}(n)$. $\qquad\square$

## 4.1 Hardness results for Policy Regret and Ranking Regret

It is easy to see that our technique for proving hardness easily extends to ranking regret (and therefore, policy regret). The reduction simply uses any algorithm for minimizing ranking regret in Algorithm 1 as $\mathsf{Alg}_{\mathsf{osco}}$. This is because in the proof of Theorem 1, the set $\mathcal{D}_\phi$ has the property that exactly one action $V_t \in \mathcal{D}_\phi$ is awake in any round $t$, and $\ell_t(V_t) = \mathbf{1}[y_t \neq \widehat{y}_t]$. Thus, if we consider a ranking where the actions in $\mathcal{D}_\phi$ are ranked at the top positions (in arbitrary order), the loss of this ranking exactly equals the number of errors made by the disjunction $\phi$ on the input sequence. The same arguments as in the proof of Theorem 1 then imply that the regret of $\mathsf{Alg}_{\mathsf{disj}}$ is bounded by that of $\mathsf{Alg}_{\mathsf{osco}}$, implying the hardness result.

# 5 Hard Instances for Specific Problems

Now we apply Theorem 1 to prove that many online sleeping combinatorial optimization problems are as hard as PAC learning DNF expressions by constructing hard instances for them. Note that all these problems admit efficient no-regret algorithms in the non-sleeping setting.

## 5.1 Online Shortest Path Problem

In the ONLINE SHORTEST PATH problem, the learner is given a directed graph $G = (V, E)$ and designated source and sink vertices $s$ and $t$. The ground set is the set of edges, i.e. $U = E$, and the decision set $\mathcal{D}$ is the set of all paths from $s$ to $t$. The sleeping version of this problem has been called the ONLINE SABOTAGED SHORTEST PATH problem by Koolen et al. [2015], who posed the open question of whether it admits an efficient no-regret algorithm. For any $n \in \mathbb{N}$, a hard instance is the graph $G^{(n)}$ shown in Figure 2. It has $3n + 2$ edges that are labeled by the elements of ground set $U = \bigcup_{i=1}^{n} \{(i, 0), (i, 1), (i, \star)\} \cup \{0, 1\}$, as required. Now note that any $s$-$t$ path in this graph has length exactly $n + 1$, so $\mathcal{D}$ satisfies the heaviness property. Furthermore, the richness property is clearly satisfied, since for any $s \in \{0, 1, \star\}^n \times \{0, 1\}$, the set of edges $\{(1, s_1), (2, s_2), \dots, (n, s_n), s_{n+1}\}$ is an $s$-$t$ path and therefore in $\mathcal{D}$.

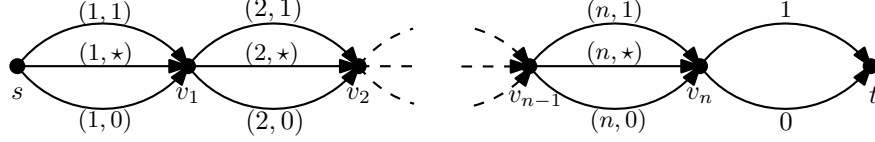

Figure 2: Graph $G^{(n)}$.

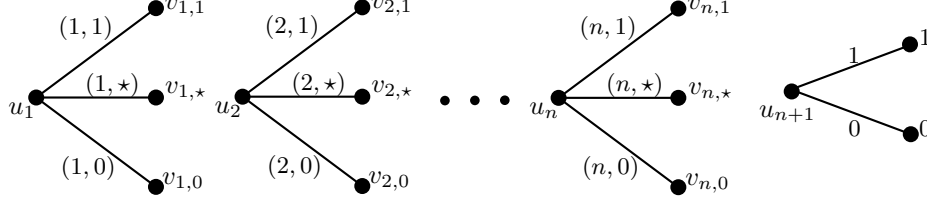

Figure 3: Graph $P^{(n)}$. This is a complete bipartite graph as described in the text, but only the special labeled edges shown for clarity.

## 5.2 Online Minimum Spanning Tree Problem

In the ONLINE MINIMUM SPANNING TREE problem, the learner is given a fixed graph $G = (V, E)$. The ground set here is the set of edges, i.e. $U = E$, and the decision set $\mathcal{D}$ is the set of spanning trees in the graph. For any $n \in \mathbb{N}$, a hard instance is the same graph $G^{(n)}$ shown in Figure 2, except that the edges are undirected. Note that the spanning trees in $G^{(n)}$ are exactly the paths from $s$ to $t$. The hardness of this problem immediately follows from the hardness of the ONLINE SHORTEST PATHS problem.

## 5.3 Online $k$-Subsets Problem

In the ONLINE $k$-SUBSETS problem, the learner is given a fixed ground set of elements $U$. The decision set $\mathcal{D}$ is the set of subsets of $U$ of size $k$. For any $n \in \mathbb{N}$, we construct a hard instance with parameter $n$ of the ONLINE $k$-SUBSETS problem with $k = n + 1$ and $d = 3n + 2$. The set $\mathcal{D}$ of all subsets of size $k = n + 1$ of a ground set $U$ of size $d = 3n + 2$ clearly satisfies both the heaviness and richness properties.

## 5.4 Online $k$-Truncated Permutations Problem

In the ONLINE $k$-TRUNCATED PERMUTATIONS problem (also called the ONLINE $k$-RANKING problem), the learner is given a complete bipartite graph with $k$ nodes on one side and $m \geq k$ nodes on the other, and the ground set $U$ is the set of all edges; thus $d = km$. The decision set $\mathcal{D}$ is the set of all maximal matchings, which can be interpreted as truncated permutations of $k$ out of $m$ objects. For any $n \in \mathbb{N}$, we construct a hard instance with parameter $n$ of the ONLINE $k$-TRUNCATED PERMUTATIONS problem with $k = n + 1$, $m = 3n + 2$ and $d = km = (n + 1)(3n + 2)$. Let $L = \{u_1, u_2, \ldots, u_{n+1}\}$ be the nodes on the left side of the bipartite graph, and since $m = 3n + 2$, let $R = \{v_{i,0}, v_{i,1}, v_{i,\star} \mid i = 1, 2, \ldots, n\} \cup \{v_0, v_1\}$ denote the nodes on the right side of the graph. The ground set $U$ consists of all $d = km = (n + 1)(3n + 2)$ edges joining nodes in $L$ to nodes in $R$. We now specify the special $3n + 2$ elements of the ground set $U$: for $i = 1, 2, \ldots, n$, label the edges $(u_i, v_{i,0}), (u_i, v_{i,1}), (u_i, v_{i,\star})$ by $(i, 0), (i, 1), (i, \star)$ respectively. Finally, label the edges $(u_{n+1}, v_0), (u_{n+1}, v_1)$ by 0 and 1 respectively. The resulting bipartite graph $P^{(n)}$ is shown in Figure 3, where only the special labeled edges are shown for clarity.

Now note that any maximal matching in this graph has exactly $n+1$ edges, so the heaviness condition is satisfied. Furthermore, the richness property is satisfied, since for any $s \in \{0, 1, \star\}^n \times \{0, 1\}$, the set of edges $\{(1, s_1), (2, s_2), \ldots, (n, s_n), s_{n+1}\}$ is a maximal matching and therefore in $\mathcal{D}$.

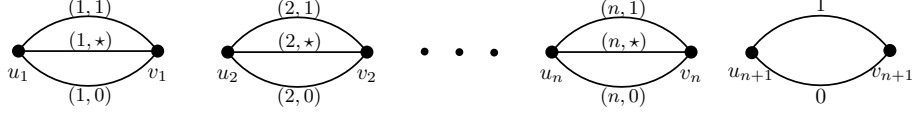

Figure 4: Graph $M^{(n)}$ for the ONLINE BIPARTITE MATCHING problem.

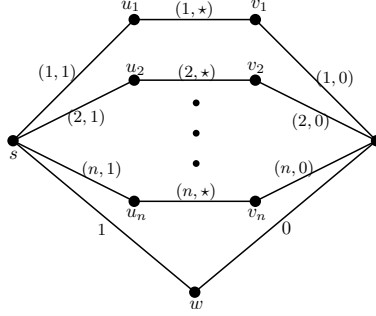

Figure 5: Graph $C^{(n)}$ for the ONLINE MINIMUM CUT problem.

## 5.5 Online Bipartite Matching Problem

In the ONLINE BIPARTITE MATCHING problem, the learner is given a fixed bipartite graph $G = (V, E)$. The ground set here is the set of edges, i.e. $U = E$, and the decision set $\mathcal{D}$ is the set of maximal matchings in $G$. For any $n \in \mathbb{N}$, a hard instance with parameter $n$ is the graph $M^{(n)}$ shown in Figure 4. It has $3n + 2$ edges that are labeled by the elements of ground set $U = \bigcup_{i=1}^{n}\{(i,0),(i,1),(i,\star)\} \cup \{0,1\}$, as required. Now note that any maximal matching in this graph has size exactly $n + 1$, so $\mathcal{D}$ satisfies the heaviness property. Furthermore, the richness property is clearly satisfied, since for any $s \in \{0,1,\star\}^n \times \{0,1\}$, the set of edges $\{(1,s_1),(2,s_2),\ldots,(n,s_n),s_{n+1}\}$ is a maximal matching and therefore in $\mathcal{D}$.

## 5.6 Online Minimum Cut Problem

In the ONLINE MINIMUM CUT problem the learner is given a fixed graph $G = (V, E)$ with a designated pair of vertices $s$ and $t$. The ground set here is the set of edges, i.e. $U = E$, and the decision set $\mathcal{D}$ is the set of cuts separating $s$ and $t$: a cut here is a set of edges that when removed from the graph disconnects $s$ from $t$. For any $n \in \mathbb{N}$, a hard instance is the graph $C^{(n)}$ shown in Figure 5. It has $3n + 2$ edges that are labeled by the elements of ground set $U = \bigcup_{i=1}^{n}\{(i,0),(i,1),(i,\star)\} \cup \{0,1\}$, as required. Now note that any cut in this graph has size at least $n + 1$, so $\mathcal{D}$ satisfies the heaviness property. Furthermore, the richness property is clearly satisfied, since for any $s \in \{0,1,\star\}^n \times \{0,1\}$, the set of edges $\{(1,s_1),(2,s_2),\ldots,(n,s_n),s_{n+1}\}$ is a cut and therefore in $\mathcal{D}$.

## 6 Conclusion

In this paper we showed that obtaining an efficient no-regret algorithm for sleeping versions of several natural online combinatorial optimization problems is as hard as efficiently PAC learning DNF expressions, a long-standing open problem. Our reduction technique requires only very modest conditions for hard instances of the problem of interest, and in fact is considerably more flexible than the specific form presented in this paper. We believe that almost any natural combinatorial optimization problem that includes instances with exponentially many solutions will be a hard problem in its online sleeping variant. Furthermore, our hardness result is via stochastic i.i.d. availabilities and losses, a rather benign form of adversary. This suggests that obtaining sublinear per-action regret is perhaps a rather hard objective, and suggests that to obtain efficient algorithms we might need to either (a) make suitable simplifications of the regret criterion or (b) restrict the adversary's power.

## Footnotes

[3]In this paper, we use the poly$(\cdot)$ notation to indicate a polynomially bounded function of the arguments.

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
