[Reviews · NeurIPS 2016]

Reviewer 1

Summary

This paper shows a general reduction that can be use to show hardness results for a large class of problems, of which one archetypal representative is the on-line shortest path problem when at any time step, any subset of edges may be unavailable. The reduction shows that the problems are at least as hard as the notoriously open problem of PAC-learning DNF formulas, while the versions in which all actions are always available is known to be efficiently solvable.

Qualitative Assessment

The topic of the paper is timely and interesting. Combinatorial experts in various forms are an active area of research that offers nice connections between learning theory and classical combinatorial algorithmics. This is a very nice theory paper that introduces a general reduction scheme and then uses it to provide hardness results for several familiar problems. The impact of the contribution is maybe somewhat lessened by the fact that hardness results for these problems are already known for more stringent notions of regret. The present paper extends those results to per-action regret. However, this is not just minor tinkering, since this notion of regret is veryu natural, and one might even think it is interesting by itself to find natural examples that demonstrate a separation between different notions of regret. The paper is very well written, with clear and consise proofs for all claims. I couldn't find any suggestions for improvement on that score.

Confidence in this Review

2-Confident (read it all; understood it all reasonably well)


Reviewer 2

Summary

This paper shows that for many combinatorial problems (shortest path, minimum spanning tree, minimum k-subset, minimum cut, bipartite matching, etc.), achieving sublinear per-action regret efficiently in the adversarial online sleeping setting is as least as hard as learning DNF efficiently in the PAC learning setting. This remains true also when losses and sets of available action elements are drawn i.i.d. from a joint arbitrary distribution. The result is obtained by constructing a simple combinatorial property of a set D of actions allowing to reduce the problem of online agnostic learning of disjunctions (a problem known to be as hard as PAC learning of DNF) to the online sleeping combinatorial problem on D. The paper then proves that this combinatorial property is shared by many natural sets D and, in doing so, it solves a COLT 2015 open problem.

Qualitative Assessment

A very nice result with a surprisingly clean proof. The combinatorial property provides novel crucial insights on the problem. The paper is well written and well placed in the context of related works. The result heavily relies on a stream of previous results, but I believe that the net contribution is still significant. I do not have specific complaints on the paper. In terms of what to do next I think it would be interesting to investigate concrete examples of combinatorial classes with exponentially many actions admitting efficient low per-action regret algorithms (if they exist).

Confidence in this Review

2-Confident (read it all; understood it all reasonably well)


Reviewer 3

Summary

The paper studies the hardness of achieving low regrete for the combinatorial sleeping experts problem (whose special case is the online shortest path problem) where at each round an action needs to be chosen that corresponds to a subset of elements from a universe (for online shortest path this would be the edges on a path from all edges). Further some set of elements may be forbidden at each time step. By reduction from agnostic learning of Disjunctions, the paper shows that this problem cannot have a regret O(poly(size of universe)T^{1-delta}) for any positive delta. Hardness for other online problems such as online minimum spanning Tree, online min cut also follow from the reduction.

Qualitative Assessment

There is a large number of variants of the sleeping experts problem that have been studied the past with several types of loss functions such as per-action regret, policy regret and ranking regret. In this large collection this paper feels somewhat incremental for this conference. The presentation can be improved. Since not every reader may be familiar with the sleeping experts setting, you may want to present a motivating application even if there is prior art for this problem -- this would make is easier for the reader to appreciate the formal objective. There is a large number of definitions before one gets to the problem specification. In the reduction from learning disjunction in section 4 you just state the reduction formally without presenting the high level idea. Please explain the main ideas informally first. Minor errors: line 152, fix "we identify satisty two.."

Confidence in this Review

2-Confident (read it all; understood it all reasonably well)


Reviewer 4

Summary

The paper considers online combinatorial optimization with potentially unavailable (i.e., "sleeping") components in the various rounds. The main result of the paper shows via a reduction that several variants of the problem are as hard as PAC learning DNF formulas, one of the central (unresolved) problem of learning theory. This result can be considered as a strengthening of the hardness result by Kanade and Steinke, and resolves an open problem proposed by Koolen et al. in 2015.

Qualitative Assessment

The paper considers online combinatorial optimization with potentially unavailable (i.e., "sleeping") components in the various rounds. The main result of the paper shows via a reduction that several variants of the problem are as hard as PAC learning DNF formulas, one of the central (unresolved) problem of learning theory. This result can be considered as a strengthening of the hardness result by Kanade and Steinke, and resolves an open problem proposed by Koolen et al. in 2015. The motivation of the problem is clear, and the whole paper is easy to follow and readily polished (maybe a table of the known result could be a little helpful for the reader), with the reduction presented in an attractively simple (but not at all trivial) form. The result fully resolves the open problem proposed by Koolen et al. in 2015 (accepting that PAC learning DNF formulas are hard), showing that the problem formulated there is intractable. This, as the authors remark at the end of the paper, urges us to come up with relaxations of the original problem that render it tractable but that do not require complete independence between the rewards and the indices describing when the arms sleep. Some further remarks: - "We identify satisfy" in line 152 seems strange. - Does V in line 191 denote the awake action in D_\phi?

Confidence in this Review

2-Confident (read it all; understood it all reasonably well)


Reviewer 5

Summary

The paper solves an open problem presented at COLT 2015 [Koolen et. al. 2015] and proves the hardness of Online Sabotaged Shortest Path problem with per-action regret, and furthermore, establishes a hardness results for a general category of Online Sleeping Combinatorial Optimization (OSCO) Problems. It defines OSCO problems as online combinatorial optimization problems in sleeping setting in which in each trial a subset of actions become unavailable due to unavailability of a subset of elements. Basically, the paper shows that OSCO problems are as hard as PAC-Learning of DNF expressions -- which is a long standing open problem. In a nutshell, the hardness result is shown as follows: the paper reduces the problem of "Online Agnostic Learnign of Disjuction" into "Hard Instances" of OSCO problems with per-action regret via a proof that has similarities to a proof in [Kanade and Steinke, 2014]. Incorporating the results of related works like [Kanade and Steinke, 2014], [Kearns et. al. 1994], [Kalai et. al. 2012], the paper mentions that "Online Agnostic Learning of Disjuction" is at least as hard as "Agnostic Improper PAC Learning of Disjuctions" which is also at least as hard as "PAC-Learning of DNF expression". Interestingly enough, because of the online-to-batch conversion argument in [Kanade and Steinke, 2014], the hardness results seem to be also true for a benign form of adversary namely stochastic availabilities and loss (i.e. they are drawn from an unknown but fixed joint distribution). After proving the general hardness results, the paper provides "Hard Instances" of various well-known OSCO problems to establish their hardness in particularv -- including Online Sabotaged Shortest Path.

Qualitative Assessment

The main proof of the general hardness result is clearly explained and looks sound and correct. The idea of the proof is similar to [Kanade, Steike 2014] and therefore not quite ground-breaking, but notably novel. Other technicalities of the paper are also well written to some extent; however, there are still some confusing parts. Concretely, section 3 is not straight-forward to follow. For example, in the first paragraph the difference between "Agnostic Improper PAC-learning of disjuction " vs "PAC-learning of DNF Expressions" is not clear -- perhaps mainly because there is a discrepency between the naming in the paper and the related work [Kalai et. al. 2012] in which the term "Reliable Agnostic Learning" is used instead. Also the reviewer could not see the hardness result even in iid case using [Kanade Steinke 2014] (it does not seem to be clearly mentioned in cited paper). Generally, the paper has a good flow and it is well-organized. The presentation is quite readable and understandable. The reviewer believes the impact is significat since it not only solves an open problem presented in COLT 2015, but also establishes a more general hardness for a category of problems with a loose form of adversarial setting.

Confidence in this Review

2-Confident (read it all; understood it all reasonably well)